# Telling Our Stories: Resilience during Resettlement for African Skilled Migrants in Australia

**DOI:** 10.3390/ijerph18083954

**Published:** 2021-04-09

**Authors:** Lillian Mwanri, Leticia Anderson, Kathomi Gatwiri

**Affiliations:** 1College of Medicine and Public Health, Flinders University, Adelaide 5042, Australia; 2Faculty of Arts, Business and Law, Southern Cross University, Lismore 2480, Australia; leticia.anderson@scu.edu.au; 3Centre for Children & Young People, Faculty of Health, Southern Cross University, Gold Coast 4225, Australia; kathomi.gatwiri@scu.edu.au

**Keywords:** African migrants, resilience, race, Australia

## Abstract

Background: Emigration to Australia by people from Africa has grown steadily in the past two decades, with skilled migration an increasingly significant component of migration streams. Challenges to resettlement in Australia by African migrants have been identified, including difficulties securing employment, experiences of racism, discrimination and social isolation. These challenges can negatively impact resettlement outcomes, including health and wellbeing. There has been limited research that has examined protective and resilience factors that help highly skilled African migrants mitigate the aforementioned challenges in Australia. This paper discusses how individual and community resilience factors supported successful resettlement Africans in Australia. The paper is contextualised within a larger study which sought to investigate how belonging and identity inform Afrodiasporic experiences of Africans in Australia. Methods: A qualitative inquiry was conducted with twenty-seven (n = 27) skilled African migrants based in South Australia, using face-to-face semi-structured interviews. Participants were not directly questioned about ‘resilience,’ but were encouraged to reflect critically on how they navigated the transition to living in Australia, and to identify factors that facilitated a successful resettlement. Results: The study findings revealed a mixture of settlement experiences for participants. Resettlement challenges were observed as barriers to fully meeting expectations of emigration. However, there were significant protective factors reported that supported resilience, including participants’ capacities for excellence and willingness to work hard; the social capital vested in community and family support networks; and African religious and cultural values and traditions. Many participants emphasised their pride in their contributions to Australian society as well as their desire to contribute to changing narratives of what it means to be African in Australia. Conclusions: The findings demonstrate that despite challenges, skilled African migrants’ resilience, ambition and determination were significant enablers to a healthy resettlement in Australia, contributing effectively to social, economic and cultural expectations, and subsequently meeting most of their own migration intentions. These findings suggest that resilience factors identified in the study are key elements of integration.

## 1. Introduction

As an economically prosperous nation, Australia has a long history of migration, and continues to be an attractive destination country for migrants. Technological developments and increased ease of communication and mobility have enabled a wider variety and number of people, including those from African nations, to relocate and settle in Australia. The increases in ease of movement and global migration have led to changes in sociodemographic dynamics and the makeup of societies and communities across many nations [1,2,3]. Through much of its history, Australia has invited migrants from across the world to resettle and build the nation [3,4,5], and today, nearly 30% of the resident population were born overseas [6]. The migration of Africans to Australia diversifies the groups of migrants who come to resettle and seek opportunities in this country. Understanding how migrants adapt and acculturate within destination countries post-migration is an emerging field of research with significant implications for policy and healthy resettlement [7,8].

Although population migration creates opportunities, it is also known to pose a variety of challenges for both the migrating and host communities [1,9], making it necessary to conduct research studies to inform policies and practices for regularly evolving situations. Prior research on migration in relation to African diasporic experiences has been associated with deficit-focussed approaches that portray African migrants as a threat and a liability [10]. International research in this field has suggested that ‘migration research could benefit from using a strengths-based approach, such as resilience, in understanding the experiences of migrants’ [11].

Migration to Australia among Africans has increased in recent years due, in part, to Australian Government humanitarian migration policies towards Africa [5,12], and in part due to policies designed to attract skilled migrants with experience in areas where there is a shortage of skills through its general skilled migration program. The skilled migration program recorded the highest numbers of skilled migrants to Australian during the year 2004–2005, with skilled migrants granted visa accounting for approximately 60 per cent of the entire Australian migration program in that year [13]. Despite the increasing migration to Australia of African people, particularly African skilled migrants, there is little research depicting their resilience. Resilience can be broadly ‘conceptualised as the ability to overcome life challenges and transform such challenges into positive growth’ [11]. Resilience is an important aspect of life and necessary for human existence and survival. Understanding more about skilled African migrants’ resilience provides evidence and a significant resource to inform policies and practices that can support the health and wellbeing of these populations and their Australian host community’s prosperity. As part of a larger study aimed at exploring the complexities of belonging, and the dynamics of change that skilled African migrants face after relocating to Australia [14], the current paper describes the mechanisms of coping and resilience factors demonstrated among this cohort, which are enablers to their effective and healthy resettlement.

### Social Resilience and Afrocentrism as Theoretical Frameworks

As a framework, social resilience [15] is understood as the ability of community groups or communities to withstand external shocks and stressors without significant disruption of their social fabric. Social resilience comprises community dynamics and processes of positive adaptation when facing significant adversity [15,16]. There are varying perspectives on what community *is*, but in the context of this paper, ‘community’ is defined as a group of people who share common value systems, have major common needs, share interests and have similar or shared experiences and identities [17,18]. Community is known to provide a space within which members develop a sense of attachment while engaging in networks that function to cushion and support them to ‘bounce back’ from adverse experiences [17]. Characteristics of community structures and interactions have been identified as complex, but overall, members of shared communities share common traits that build resilience through ideas, experiences, skills and knowledge [18]. These characteristics have been reported to assist individuals, families and communities to overcome shocks and stresses, including changes in government policy, civil strife, or environmental hazards and resources [19]. For skilled African migrants in Australia, social communities and communities of attachment, where a sense ‘feeling at home’ is inculcated, can provide the foundation for a successful new life in Australia. The importance of using a strength-based approach such as social resilience in understanding the experiences of migrants has been recommended to improve the knowledge about how communities deal with adversities [19], or major life changing challenges, which migration to and resettlement in new countries constitute.

In addition to the everyday challenges of resettlement which would be anticipated for any migrant to a new socio-cultural setting, Black African migrants face additional obstacles to resettlement in an environment where race has particular salience. Within the Australian context, Black Africans are ‘marked’ as different from the white, Anglo-Saxon heritage majority through a combination of ‘visibilities’ including race, dress, and accent [20]. Black Africans in Australia are therefore ‘hypervisible’ and are constructed as perpetually outside the boundaries of mainstream normative conceptions of Australian identity. These factors can therefore contribute to overscrutinisation and marginalisation of members of Black African communities [20].

Resilience frameworks in migration discourses, particularly those that theorise experiences of Black migrants, need to employ strength-based and non-deficit approaches, while also acknowledging the additional challenges that resettlement in predominantly white contexts present. We also assert that there is a corresponding need for culturally affirming theoretical frameworks and research methodologies that recognise the cultural strengths of migrating communities. We contend that this need can be addressed through the utilisation of Afrocentricity to investigate Afrodiasporic experiences. Afrocentric epistemologies, applied appropriately, can offer a powerful alternative to and critique of Eurocentric perspectives and discourses on resilience [21]. Utilising paradigms that privilege African ways of knowing, being and doing to solve human and social problems is a valid form of interpreting social and psychological issues affecting Africans in order ‘to create relevant approaches of personal, family, and community healing and societal change’ [22]. Afrocentric-informed research offers an innovative approach to exploring challenges for Afrodiasporic communities in Australia in that it identifies and utilises the community’s knowledge, resilience, and expertise to inform knowledge and design its own solutions.

*Sancofa* and *Ubuntu* are the principle Afrocentric philosophies that inform our analysis. *Sancofa* as a framework represents the embodiment of a mythological bird that flies forward but with its head turned backward, symbolising the Ghanaian Akan proverb that, *‘it is not wrong to go back for that which you have forgotten*’ [14]. This Afrocentric philosophy acknowledges the importance of *returning to* and renewing African knowledge and experiences that have been marginalised and/or forgotten. Invoking it in our analysis helps to highlight how the process of *returning to self* can facilitate resilience and successful resettlement. *Ubuntu* on the other hand is centred on the premise that, *‘I am because we are, and since we are, therefore I am’. Ubuntu* is a philosophical framework that argues that we are *made human* through the process of humanising others [23,24], such that the willingness to *see*, *feel and enter* the depth of other people’s experiences through a humane process produces interconnectedness and change. Supplementing the use of a social resilience framework, we also apply the philosophies of *Sancofa* and *Ubuntu* in framing and interpreting the responses of participants in this research, as this enables us to prioritise collectivist and group identity values that significantly advance the research aim and help ensure that conclusions emerging from the study are informed by culturally appropriate knowledge.

## 2. Methodology

The study methods and reporting were guided by consolidated criteria for reporting a qualitative study (COREQ) checklist [25]. This checklist comprises 32 items within three domains including: (i) Research team and reflexivity, (ii) Study design, and (iii) Data, collection analysis and findings.

### 2.1. Research Team and Reflexivity

The authors are senior academics from two universities, skilled and experienced in qualitative research methodologies. Two of the authors (LM and KG) are African Migrants and have extensive networks within the African communities across Australia. While our involvement could impact the participants’ responses, care was undertaken to minimise conscious bias. However, we also acknowledge that our knowledge and insight as ‘insiders’ would be beneficial and enriching an understanding of migration and the experiences of re-settling in Australia [26].

### 2.2. Study Design and Data Collection

The study design employed a qualitative method of inquiry, which presents a unique opportunity for exploring participants’ lived experiences of resettlement. This approach focuses on the meaning and interpretation of the respondents’ subjective experiences and how these meanings are connected to their broader experiences of inclusion and belonging.

The study was conducted in Adelaide, South Australia. According to the 2016 Australian Census, Adelaide had a population of approximately 1.3 million residents. Similar to most urban settings in Australia, most residents of Adelaide have Anglo-Australian heritage indicating that Black Africans are part of a culturally and linguistically diverse minority population group [5].

The researchers recruited participants through existing connections, community networks and snowball sampling. In total, twenty-seven (n = 27) participants including 15 men and 12 women from sub-Saharan African countries who had migrated to Australia as skilled migrants were interviewed. Participants number and countries of origin included: Fifteen from Kenya, three from Nigeria, two each, respectively from Zambia, Tanzania and Ghana and one each, respectively from Zimbabwe, South Africa and Rwanda. Most participants were middle class working professionals. Participants’ occupations included senior roles in the fields of banking and finance, business, medicine and health, social work, engineering, mining and academia. Participants’ length of stay in Australia varied and ranged between two and twenty-five years.

Interviews were conducted by two members of the research team (KG and LM), academics from within the African community in Australia. Interviews were conducted in the participants’ place of choice which included homes, offices or other mutually agreed places and lasted approximately between 45–90 min.

### 2.3. Data Analysis and Ethical Consideration

Individual interviews were recorded digitally, transcribed and analysed deductively and inductively using a thematic framework analysis [26,27]. Prior to the interviews, each participant was informed about the purpose of the study, the voluntary nature of participation, and their right to withdraw their participation at any time, without consequence. Before commencing interviews, the researchers ensured that informed consent was obtained. Participants were assured that the data or information that they provided during the interview was confidential and unidentifiable. This study was approved by the Southern Cross University Ethics Committee (project number ECN-18-002).

## 3. Findings

Despite a mixture of resettlement and environmental barriers, including limitations in employment opportunities, experiences of discrimination and subtle racism, participants demonstrated strong resilience and the ability to cope with challenges that they encountered during their journey to re-establish themselves in the new host nation. The resilience factors identified included: (1) capacities for excellence, that included willingness to take any available opportunities to achieve their goals; (2) social capital through community and family support networks; (3) strongly held African and religious values (4); pride in contributing to Australian society; and (5) desire to change the narratives about Africans in Australia. These themes are presented and discussed further below.

### 3.1. Willingness to Work Outside of Existing Expertise

A significant theme referred to by several participants was the exceptional work ethic and capacity for professional excellence among African migrants. Despite the challenges in securing matching (skills, education, ability, and expectations) job opportunities, this was partially mitigated by their willingness and readiness to work hard and take any available opportunities. One participant, Jimmy, stated, ‘*Africans will do anything, they will figure out anything that they can do–and do it, and they will work as hard as possible, they will pick fruit from the trees if they have to’*. Mukisa also noted that despite being based in a specialist urban medical centre, ‘*I still go back and cover doctors who are on leave in the rural clinics and I work hard’.* African migrants were keen to show how hard they were willing to work to obtain a better life for themselves and their families. Strong work ethics, hard work, determination and perseverance were described as important in obtaining and sustaining job opportunities. For example, Amani noted how she was the only candidate from her workplace who received an ongoing role after a round of interviews. The feedback she received from the hiring managers was that she demonstrated an excellent work ethic through her extensive preparation for the interview, and this was the reason she got the permanent role. She also noted that her ‘persistence’ in taking the steps toward promotion led to more senior roles, even though many of her ‘Caucasian’ colleagues ‘gave up along the way’, because they felt the process was ‘too strenuous’. She added, ‘*I had talked to* [African] *friends … who have been in even more senior positions* [in Australia] *than they had been, in their own countries, and going into skills or jobs that they have never done before*”.

Participants emphasised that Africans in Australia ‘just wanted a chance’, and if given the opportunities they deserved, they could make even more significant and positive contributions to Australia society. Wanjiru exemplifies this by stating:
*You know, we have hard working people. We just want what everyone else wants, what most people want. Just give us a chance. We want what everyone else want, best for our kids, to better our lives…I want when someone sees me, they [see a] hard working woman who has travelled far away to come and work hard for her family and make a better life for herself and for her family members*.(Wanjiru)

A theme that was less explicitly emphasised was the observation that because non-migrant Australians already had existing social capital and networks, and African migrants often did not have this benefit, they could not rely on what Awinja described as waiting for ‘*someone else to come and … fish you out’* for opportunities. Instead, she emphasised that for African skilled migrants, *‘it is really for us to position ourselves really well and advocate and become our own self advocates*’.

### 3.2. Social Capital Through Community and Family Support Networks

Family and community support networks were strongly highlighted by participants as one of the factors in building resilience and perseverance while managing the challenges of migration and resettlement in Australia. The earlier arrived African families who were not related or previously known to each other, acted as extended families to form a supportive network for newer arrivals. John and Julia valued how the community facilitated their resettlement and noted, ‘*right the day we came in’*, when the African families who had already settled in Australia welcomed them at dinner and social events. This network of African families, not previously known to the participants, provided extensive support for this family over the first critical three months of arriving in Australia. They reflected:
*That support saw us through and I think that is a very important point to put across. The family literally took us by [our] hands and they were in our house literally every day for the next three months. If there was an activity in Adelaide, they came and picked us. If we were going shopping, they came and picked us. If there was visiting anyone, they came and picked us. So in three months, we had met so many people [Africans] and that kind of things made it very easy and comfortable for us*. (John and Julia)
Kissa also reflected on a similar experience:
*There was a small African community in [town in South Australia], they came around, they supported. I would say so far I had an amazing journey, basically because I met amazing people, people that have extended warmth, love, and support and understanding*. (Kissa)
Some participants provided examples of how family members from their country of origin travelled to Australia to provide essential support at critical times. Wanjiru, for example, discussed how necessary such support was when she was a new mother and studying full time:
*Our family gave us much support during that time when we were going to nursing school. My sister came from* [country name] *and she lived with us for a whole year while we were going through school. Our babies were so young then, so she helped with babysitting, and without that kind of help, oh my goodness, I don’t know how we would have pulled through the nursing school! So we cannot forget the family support*. (Wanjiru)
Within family units that had migrated together, new levels of cooperation and support were also required to achieve a successful migration experience. Banji, for example, noted that ‘*I have a very supportive spouse and so we tend, you know, to help one another’*. Wanjiru and Paul similarly emphasised the value of support within their family unit. They described how as ‘*a family we really, really, really had to learn how to work together. We form*[ed] *our own identity–so to speak–as a family, to help us cope with the challenges that we thought of or felt’*.

In addition to family support, formal organisations were also identified as a significant source of support, as most participants were members of one or more community organisations. Some participants particularly singled out the role of such associations in building resilience and experiencing belonging in the new country. This included regional or country-specific associations, faith-based organisations and church congregations and to a lesser extent, government-initiated settlement programs. For some, work colleagues, especially those who were introduced immediately upon arrival or who eventually became friends, provided a supportive structure for their resettlement and enabled the building of resilience and a sense of community. Kissa, for example, mentioned that at her first workplace immediately after arriving in Australia, ‘*there were some amazing people there that made me welcome, I went to their home to eat, they would invite me, others will invite me over with my little son, almost every weekend to come and spend time with them*’.

Some participants described how at the time of migrating, the South Australian Government’s settlement program for skilled migrants was highly supportive. Initial, formal government support was critical in building participants’ resilience and providing a springboard for their new life in Australia to take off:
*The South Australian Government program back then really looked after new skilled migrants and helped them settle. They organised a house at subsidised rent for us for the first three months. We had a two-bedroom house, which was good for us. The house was next to the school where our kids were going, so we did not have to walk far. It was next to a tramline, so we did not have to bother about having a car initially, so the entire program was a really good one…I think our great experience is one that is highly supported by the program. Without the program, I think we will be talking about totally different things altogether*. (John and Julia)

Although they did not go into further detail, the emphasis on helpful government support for skilled migrants being provided ‘back then’, suggests an assumption that this type of assistance was not necessarily available to contemporary migrants. This could then lead to more resettlement challenges.

### 3.3. Religion and Faith as Protective Factors

Spirituality and faith were important determinants that provided mechanisms to cope with challenges for many of the participants. Religion is a well recognised factor that functions both as an intrinsic and extrinsic marker of individual and population resilience [28,29,30,31]. One participant concurred with these assertions stating that, ‘*my family gets support–and a lot of it–from our church*’. Other participants reported that going to church was an important ritual for them as it enabled them to set one day off work each week to attend community service, providing them with an opportunity to connect and build friendships with others with similar values. For Banji, religious faith acted as a cushion for his family during their resettlement years:
*Having good support at church are a crucial factor that have helped in terms of settling down. So we were very strong in terms of being involved in an active church…This really helped in the way we have settled down in Australia … I guess you know those three factors–work network, church network and also network of friends–have really helped in settling down*.(Banji)
Jimmy also supported religion as protective element in these communities when he stated, ‘*most Africans have a Christian or another form of faith–and that’s where they find support’,* implying, like Amani and Banji, that religion plays a significant role as a coping tool, and is utilised as a strong resilience mechanism in overcoming adversities.

### 3.4. Pride in Contributions to Australia as Africans

Participants emphasised pride in their contributions to Australian society. This pride was linked to their contributions in professional excellence, expertise and economic contributions through dutiful payment of taxes. Other participants nominated specific aspects of their area of work expertise as advancing Australian society, which they were justifiably proud of. Sally, for example, nominated her studies and the specific scientific research she had conducted in Australia as having made a significant scientific contribution. Mukisa, a GP with a special interest in skin cancer and family medicine, noted that ‘*I worked hard…providing my skill to Australians and making sure that whatever I do, I do it at a high-level of skill and I think that has been the biggest contribution’*.

Other participants spoke with pride of their success in advocating for more diverse and culturally aware perspectives and policies:
*I can claim that I have done [a lot] to advocate for cultural issues being prioritised. I have managed to move the government away from sideling some communities, who have dealt with issues that needed certain support to be able to move up. Groups which have less resources, less networks, more disadvantage compared to the main stream, can now have more support and the government is ready to accept that, and we have seen so many resources coming and I am grateful for that*. (Maurice)

Paul similarly highlighted that African migrants add significant cultural value in Australia;
when we come here, we bring cultural diversity…we educate the people of this country on who we are, we expose them to other cultures because I think it is unfair when all their knowledge about us comes from National Geographic’.(Paul)
Most participants singled out how their financial and career success also benefitted Australians, especially through the payment of taxes and bolstering of the Australian economy. For example, Jimmy noted, ‘*I have been working for the last 15 years and earning good money and paying taxes* [and] *investing* [in] *property*’. Kissa, Maurice and Awinja all discussed their various economic contributions to Australia as illustrated below:
*Me and my husband* [both senior medical consultants] *generate a lot income, but we also support the government by paying taxes, which helps to actually do a lot of the other projects…*[so] *many Australians would benefit from our tax*.(Kissa)
*We have contributed financially and economically, being in real estate. We have investments in not 1, 2, 3, or 4 places. Our properties are scattered around, and we employ the real estate managers to manage the properties. In that way, they get money for managing our properties. We pay high taxes and we have helped the South Australian economy–we can say yes, we have done that*. (Maurice)

Jenny and Patrick also explicitly linked the payment of taxes to the value they added to Australia, in terms of what that sum would mean in their country of origin: ‘*Let me tell you the tax we pay between us, is enough to feed my whole village for a year. We have a right to be here*’.

Nkandu made similar emphasis in this direct connection stating, *‘the majority of Africans or migrants are working hard, they are paying their taxes and not being a burden to society’,* yet this was at odds with the dominant Australian public discourse constructing migrants primarily as welfare recipients:
*I hear Australians say all these things about migrants and think, ‘who do you think pays for all these stuff (Medicare, Centerlink)?’ It is the working people, and the big chunk of that working people are migrants. So, I think if there is a story to be told [it] is the contribution that migrants [make] to this country. It is just everyday people, waking up every day in the morning, go to work, obey the laws, they are peaceful with everybody, I think that is the greatest story to be told*.(Nkandu)
He further believed that ‘*Australians would be shocked that there are a lot of white Australians on welfare benefiting from* [African] *migrants going to work*. Participants also described how they managed to put aside the disadvantages that they faced so as to focus on their main intention for migrating–that is, ‘*to work hard, and find effective opportunities for themselves, their children, and communities’.* They were also able to demonstrate considerable flexibility, that supported the development of resilience and aided successful resettlement:
*I think it is important for us to be open to be willing to integrate and embrace the new culture that we find ourselves while not losing our own. We can learn what to take and what to reject from this culture and we can still embrace it and bring the positives from our own culture and the positives from this culture and make it better*. (Kissa)

Despite the challenges encountered in re-settling in Australia, participants reflected on their commitment to *remember* where they came from and *who* they were as Africans. Although they exemplified a deep commitment to integrate and abide by Australia’s culture, they also agreed on the importance of upholding their own culture and retaining their Africanness. The commitment to retain their Africanness in a society that covertly promotes ideologies of assimilation, aligns with the broader messaging of *Sancofa.* Sally narrates:
*I will not change being an African or anything. It comes with its challenges, but you know what, I love it. I love being an African and I hold no apologies for being an African. So, if I am here as an African, I can equally contribute to the society as anyone, so I believe in myself. I am very aware and I set my identity fully. That is the starting point. You need to believe yourself and accept your identity fully, with the accent, with the colour, with whatever, fully. It should not matter and you need to believe that you are in this for the long haul*. (Sally)

### 3.5. Reframing the Narrative of Africans in Australia

Pursuing excellence and acknowledging the challenges of migration and adapting were identified as subthemes and are further described below.

#### 3.5.1. Pursuing Excellence

Many participants indicated a desire to contribute to changing negative narratives of what it means to be an African in Australia. Further suggestions included a need for advocacy and more accurate and positive representation of African migrants and communities that challenged dominant mainstream narratives about ‘Africanness’ in Australia. As Jimmy stated:
*I would like to see the image of Africa change a bit more, I would like to see people* [not see] *Africa as a place where there is needy people, I would like the people see Africa as a potential partner, as a power house, as a place of great ideas have come from*. (Jimmy)

Much as participants called for more inclusion and a broader narrative shift in the Australian public discourse on Africans, many emphasised that African diaspora communities were required to ‘step up’ and actively work to change that narrative by ‘pursuing excellence’. Kissa reflected:
*Australia is a beautiful country. We are absolutely proud to be a part of it, but as people of colour, we should look for opportunities to add value. We should be people of excellence. We should stand for excellence and also in particular we need to start thinking aggressively about the next generation. As the first generation of African immigrants, we need to make sacrifices for the next generation and we must not miss that point, it would not be all about us. We got to think about 10, 20, and 30 years from now, where would we like to be in the Australian society*.(Kissa)

This emphasis on acting as advocates and positive role models for African communities echoes the responses from participants above who observed that African migrants come to Australia lacking networks and pre-existing social capital. As such there is a greater need for migrants to create their own opportunities, rather than waiting for their excellence to be recognised and rewarded by others.

Other participants presented similar ideas about how Africans in Australia can help change and reframe negative narratives predominantly through bolstering their own self-confidence and by not internalising any negative assumptions about Africans. Jimmy advised Africans in Australia to ‘*come here with something to offer’*. He adds, ‘if you are here as a skilled professional who is going to make a contribution to this country, see yourself as of that, do not see yourself as a poor African when you go to meetings’. Others cautioned Africans not to adopt a victim mentality, as this reduces the capacity for resilience and excellence. Amani advised:
*Let’s not pity ourselves, let’s not feel sorry for ourselves…what helps is for us to go out and explore, so it is up to us really to position ourselves. It’s not up to anybody else to do that for us…I am a senior social worker, and it doesn’t come easy … I would encourage everyone to work hard, but not pathologise ourselves and don’t, don’t join people in pathologising us*. (Amani)

Whilst supporting this position broadly, Sally observed that this does not always come easily. Although she proudly described herself, as an African migrants more generally, as being part of ‘a group of people with great tenacity, great intelligence’, she also reflected that ‘*we need to start believing in ourselves more and come together in a unified way to say that we are here to stay and we are not going anywhere, and we are a force to be reckoned with’*. She added:
*If all of us are working together … celebrating one another and helping and lending a hand to one another and just be ourselves and be proud of who we are and also just give the best of ourselves to this country because really, this is home even if it doesn’t feel like so for a long time, but in this moment, right here, right now, this is home*.(Sally)

Sally’s reflection can be understood as both a recognition of how challenging resettlement is, as well as a subtle questioning of whether individuals can foster this change on their own without an *Ubuntu* mentality. Her emphasis instead is on the need for *collective* action to achieve these goals, an observation that underscores the value of community and collectivist values.

#### 3.5.2. Acknowledging the Challenges of Migration and Adapting

Despite the challenges many had encountered, participants refused to see themselves as victims. They seemed to have chosen to look for the positive opportunities that Australia offers and opted to focus on what was necessary to achieve their goals of having a successful life in Australia. Many described the initial challenges, and how they overcame them with a focus on their family betterment:
*I felt my status become really low. My status* [in original country], *came with a lot of respect. So here, I just felt like I lost my status. But I do not know whether you call it resilience or what it is, I thought, you know what, this is a new beginning and I said, ‘This is a new beginning, because this is a new world’, and I focussed on the children*. (Awinja)
*It has been an interesting journey, it has had its ups, but it also has its downs. What I can say is the opportunities for the children here are much better. So if someone wanted to come, just have your goals set out. Just be clear on what you really want because if you are coming because you want to make a lot of money very quickly, you might be very disappointed. You have got to learn to look at those positives around. If you want your children to have a better life, then you need to clearly know that you would have to sacrifice a lot, on your part as a parent … I think that was the main reason why we decided to come, because at a certain time in your life, it stops being about you as a parent and it is about your children and if you can focus on that, then you would save yourself a lot of heartache*.(Julia)

Participants were very aware of the need to integrate within Australian society. While they seemed adamant that they did not need to fundamentally change who they were, they also argued that they *should* adapt to social norms in the new host nation:
*I think you need to integrate when you move to a country, you need to integrate with citizens of that country … Friends for me come from any group, so I integrate with everybody and I think integrating makes a big difference. Because then, you do not feel isolated. You will find if you keep to your [country of origin] community, you talk about [country name] social issues rather than connecting on the issues that are happening here…you will become homesick, you will always think about [country name]*. (Jabali)
Jabali added that in the resettlement process, ‘*what worked for us is to integrate with the local community, that made a big difference*’, thus emphasising the need to build connections with Australians from diverse backgrounds as an essential part of successful resettlement”.

An additional aspect to this theme was participants’ belief that that integration should be a two-way process, and not a unidirectional or assimilatory process [32]. This would require the host community to also adapt, by developing an understanding of some African cultural values and way of life. A two-way process, they argued, would have bi-directional benefits for both African migrants and mainstream Australian. Several participants also considered that since Australian is a settler state built on migration, they deserved to resettle as migrants because they were determined to contribute to building their new country. Sally reflects
*Africans are equally deserving to be here, we have equal rights to be here. But I think Africans need to be internally strong ... because the challenges will come. It is not about if they will come, but when they will come, and when they do come, what are you going to do with it. Will you let those challenges pull you down? Or, you are going to let those challenges even make you stronger, and more determined that you are here, and you are going to benefit from being here and also contribute*. (Sally)
The determination and optimism expressed by Sally was also mirrored by other participants who added that their excellence and deservedness to be in Australia contributed to their successful resettlement.

## 4. Discussion

People choose to migrate for various reasons including to access new employment or education and better opportunities [33,34], and to escape civic unrest or conflict [8,35]. Although there is now a significant emerging body of knowledge on African migrants in Australia, most of these studies have focused on refugees, so the focus of this study on skilled migrants addresses a significant gap in the extant literature [9,35]. Despite the significant challenges including racism [9,32], difficulty accessing employment [2], discrimination [9] intergenerational issues [8,36] and micro-aggressions [37] that seem to dominate experiences of African migrants during their resettlement, most draw positivity and optimism from religion, faith, hope, and community-oriented attitudes which appear to nurture resilience and social connections. Positivity and optimism have been identified in the literature as factors that build resilience among migrant groups [29,38].

For participants in the current study, the intention to migrate to Australia seemed to have been made mostly for the purpose of bettering the future opportunities of their children and families. As noted in their narratives described elsewhere in this paper, the emphasis on the positive contribution made to Australian society emerged as a strong theme. It is reasonable to hypothesise that these attributes are informed by collectivist value systems—as embodied in the philosophy of *Ubuntu*—but also partially as a response to dominant and harmful narratives about African migrants in Australia, which are frequently deficit focused and connected to negative racialised stereotypes [20,39]. Consistent with previous study findings and the journeys of this study’s participants, migration promotes a different ecological environment that fosters the diversity of culture, with an interweaving of different ways of life that includes changes to views, systems, values and aspirations [40,41], which are well demonstrated perspectives in this paper.

The social resilience frameworks [16,42] provide a robust understanding of the mechanisms that foster resilience through multi-layered factors. Resilience has been described not only as an individual psychological trait, but as a social phenomenon that is mediated by individuals’ cultures and social ecology [16]. The current study findings demonstrate that participants displayed significant resilience that enabled them to cope effectively with the challenges of migration in Australia, and positively contribute to the Australian socioeconomical and ecological systems. Individual resilience (personal qualities/traits, education background, skills, ambition, etc.), the availability and accessibility of resources such as community, and government support, all contributed to reducing resettlement challenges.

Supporting the observations of our participants in relation to religion, a multitude of research studies have identified religion, faith and prayer as protective factors that mitigate the adversities of migration and build resilience [42,43,44]. Whilst resilience is a psychological process, it is an outcome of social processes that exist in relationships between people, systems, institutions (such as churches), organisations and networks [45]. For our participants, the social aspects of church attendance, participation in faith-based organisations and related activities, and activities within their own and other families, provided a positive source of resilience-building through interpersonal connections. Similar observations have been made in the literature, identifying these components to be among protective factors that enable positive adjustment to migration [40,41,45].

Supplementary forms of social support provided by other families, African communities, organisations and other social networks were acknowledged as a significant source of cultural safety, fulfilment and belonging, which is consistent with African cultural norms in regard to collectivism and *Ubuntu* philosophies. [22,24]. The thematic display of social support and social networks in participants’ narratives underscored the value of *community* that was that was not necessarily based on close familial ties, but which exemplified the value of Afrocentric philosophical frameworks of *Ubuntu.* This African philosophy relies on a prioritisation of collectivist and group identity values in comparison to individualistic values within Western societies [24,46]. *Ubuntu* philosophical perspectives bolsters our understanding of the importance of supportive communities in the development of resilience, health and wellbeing in Afrodiasporic communities. The role of social support derived from families [47], communities [8], and community organisations [7], is well recognised in research as crucial in facilitating a healthy resettlement among migrants [7,9,10]. Collectivist and family-oriented values were also demonstrated through the reasons for migration, which was motivated not by personal gain, but to create a better future for their children, their communities and future generations.

It is also worth noting that, participants were both ready to integrate and align with the host community’s norms, as well as continue to maintain their African identity and cultural perspectives. These dichotomous perspectives correspond with the *Sancofa* principles where, whilst these participants looked forward to a bright future in Australia, they also needed to nurture their African heritage in Australia, thus exemplifying the notion that *‘it is not wrong to go back for that which you have forgotten’* [14]. Through these noted attributes of collectivism, flexibility and positivity, it is plausible to acknowledge that many skilled African migrants were able to develop a thoughtful appreciation of the positive aspects of both African *and* Australia cultures, and selectively and intentionally drew upon these resources in order to support their aspirations for a successful resettlement in Australia.

There was also significant trust placed by many participants into Australian colleagues and friends and the belief that in general, Australian people were ‘good’. This comfort was drawn from seeing people’s intrinsic goodness and trusting this collective ‘goodness’ would cushion them from any extreme negative experiences. For some participants, work colleagues extended positive relationships. This familiarity, trust and synergy in the workplace encouraged a sense of belonging and helped minimise the complexities of social issues of isolation [45]. Additionally, connections both within specific ethnic communities and within the broader Australian community offered practical and emotional support in building a new life. This supports the consensus in the literature on the importance of strong social ties and social capital in supporting quality health and wellbeing outcomes [48,49,50,51].

Importantly, it is necessary to re-emphasise participants’ pride in their decision to resettle in Australia, their strong professional identity and their admirable qualities. Some participants described in detail their achievements at work, their contributions to the Australian economy, their investment successes and their creation of employment and other opportunities for the wider Australian community. International literature on skilled migration has in the past focused on macro contextual issues such as brain drain, and migrants’ remittances [52], but this has left a knowledge gap about skilled migrants’ self-initiated and self-identified contributions to the *host* environment. We argue that there are benefits (socioeconomic, cultural and ecological) of successful resettlement for both new arrivals and receiving communities as demonstrated in the current study. As such, supporting migrant communities, including highly skilled new migrants, to resettle successfully is important for Australia to implement. This support is the most effective and empowering tool when it is embedded in social and political practices to help new migrants mitigate migration resettlement obstacles [51,52], including challenges in employment, [2] and inclusion [40], so as to improve their health and wellbeing [7], in their new country.

### Strengths and Limitations

The study involved highly skilled migrant participants who lived in South Australia but originated from different countries in Africa. The selection of this specific group of migrants enabled the researchers to explore the lived experience of this specific population from a resilience stance. The sampling approach and the use of qualitative method meant that research participants included those who were highly skilled (from particular different professional backgrounds providing snapshot of experiences from different workplaces), had self-initiated migration to Australia, could speak proficient English and were actively engaged in community groups or found via a snowball approach. This resulted in data that was rich and enabled deep understanding of individual resettlement stories. As such, the paper contributes to the body of knowledge regarding significant protective factors that mitigate the adversities of migration as it focused on identified mechanisms and sources of resilience for skilled African migrants in Australia.

However, interviewing people who are already connected to groups and the snowballing approach (which relies on social networks and is therefore not a random sampling approach) may have resulted in missing the perspectives of less connected people. Likewise, the scope of our research did not extend to views of the broader South Australian community, which would have given further insight into alternative perspectives of settlement and resilience.

## 5. Conclusions

The findings demonstrate that despite challenges experienced in resettling in Australia, skilled African migrants’ resilience, ambition and determination were significant enablers to a healthy resettlement, contributing effectively to socio-ecological, economic and cultural expectations, and subsequently meeting most of their own migration intentions. These findings suggest that resilience factors identified in the study are key elements of resettlement. These include personal qualities, education and skills, positive attitudes, family ties, religious and cultural values, communities of attachment, and social connectedness. This study highlights the ways in which resilience is enacted among African migrants and brings to light their capability in facing migration challenges and effectively contributing to the Australian economy, and to social and cultural structures. The importance of African migrants’ resilience and Afrocentricity to the subsequent welfare and healthy resettlement of their families into a new culture, and indeed the benefits to the new communities [53,54], cannot be overemphasised. The study provides significant information that can be used to inform knowledge about a healthy resettlement of groups of new migrants, applicable to Australia or in similar settings, and validates the supplementation of research approaches and methodologies with Afrocentric frameworks.

## Data Availability

The data used for the current study are not possible to share for ethical reasons. The Southern Cross University Ethics Committee requires that the data are not shared without the consent of participants, who are currently not accessible.

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
