# Peer review of "Telling Our Stories: Resilience during Resettlement for African Skilled Migrants in Australia"

_ijerph, 2021, doi:10.3390/ijerph18083954_

Round 1
Reviewer 1 Report
Dear Authors, you have dedicated very few attention to Australia's migration policies and programmes tailored to attract high skilled migrants.
You should dedicate some explanations in the Introduction to that: it would help you in robusting your Conclusions too.
Author Response
Thank you very much for this observation. We have added information about skilled migration policies in the introduction section on. The study adds value to the literature by addressing issues of resilience as a predictor for a successful settlement in people who are not refugees (migrated as skilled migrants) who have settled in a relatively small city a high income country. We have also made some statement in the conclusion to align with the comments made by addition information about African migration to Australia.
Reviewer 2 Report
This article’s topic is very interesting and presents some new reflections in the field of skilled migrations. The article is well organized, clear and appropriate in every section. However, I believe two suggestions may implement the efficacy of this manuscript:
1) It is useful to integrate structural variables of interviewees (age, Country of origin, educational level, reason for migration, years of presence in the host Country, earnings) in sample description, because they would allow to read the narrations in a more appropriate way. Literature also considers those variables relevant for the objectives investigated by authors. In addition if the influence of these variables is not considered in results and discussion sections, the real meaning of some narratives results simplified and obscured
2) the Sancofa perspective used in the manuscript (discussion part) is hardly traceable in the interviewees speeches and it sounds like a theoretical-inappropriate grid of the authors applied to the quotations
Self-citations should be controlled and reduced and some edit errors (lines 19 and 252) need a check.
Author Response
This article’s topic is very interesting and presents some new reflections in the field of skilled migrations. The article is well organized, clear and appropriate in every section. However, I believe two suggestions may implement the efficacy of this manuscript:
Response: Thank you for your very complementary comments. We really appreciate these.
- It is useful to integrate structural variables of interviewees (age, Country of origin, educational level, reason for migration, years of presence in the host Country, earnings) in sample description, because they would allow to read the narrations in a more appropriate way. Literature also considers those variables relevant for the objectives investigated by authors. In addition, if the influence of these variables is not considered in results and discussion sections, the real meaning of some narratives results simplified and obscured
Response 1: Thank you for this observation which has enriched our paper. We have added participants numbers for each country of origin and the average year of participants stay in Australia. We have also added the general information indication of their earning. As the African community in the research setting is very small, we have avoided providing specific details of participants because further details can lead to participants being identified public, which would be against ethics committee’s requirements. The reason for migration was highlighted before, and all migrants migrated to Australia voluntarily in search of better opportunities and to improve the their lives and to provide a better opportunity for their children in the future. We also highlighted earlier that all participants had earned at least one University degree from their country of origin. We have made correspond edition in the manuscripts’ text
- the Sancofa perspective used in the manuscript (discussion part) is hardly traceable in the interviewees speeches and it sounds like a theoretical-inappropriate grid of the authors applied to the quotations
Self-citations should be controlled and reduced and some edit errors (lines 19 and 252) need a check.
Response 2: Thank you for this question. As this is a qualitative inquiry, we have used different theories to interpret the data (narratives from participants). We would like to acknowledge that Sancofa and Ubuntu are the principle Afrocentric philosophies that informed our analysis. Sancofa as a framework represents the embodiment of a mythological bird that flies forward but with its head turned backward, symbolising the Ghanaian Akan proverb that “it is not wrong to go back for that which you have forgotten. We acknowledge that there were no traces of this theory in participants voice, but when we interpreted quotations such as, “I think it is important for us to be open to be willing to integrate and embrace the new culture that we find ourselves while not losing our own. We can learn what to take and what to reject from this culture and we can still embrace it and bring the positives from our own culture and the positives from this culture and make it better. (Kissa)
We have also added a narrativewhich would be interpreted as consistent with the Sancofa sentiment from a participant, “I will not change being an African or anything. It comes with its challenges, but you know what, I love it. I love being African and I hold no apologies for being African. So, if I am here as an African, I can equally contribute to the society as anyone, so I believe in myself. I am very aware and I set my identity fully. That is the starting point. You need to believe yourself and accept your identity fully, with the accent, with the colour, with whatever, fully. It should not matter and you need to believe that you are in this for the long haul. (Sally)”
We have also reduced self citation as much as feasible.
Reviewer 3 Report
Dear Authors,
Thank you so much for providing a manuscript on this important issue. The manuscript would benefit from the revision.
Title: Resettlement in where? The authors could consider adding more clarity by indicating setting migrants from Africa who resettled in – Region/area/setting
Methods – the authors did not mention if the study follows Consolidated criteria for reporting qualitative research (COREQ): a 32-item checklist for interviews and focus groups – guidelines. The authors are suggested to carefully check COREQ and follow its principles thoughts the manuscript.
Results –the authors may consider presenting main themes as tables.
Page 4, Line 177: the word “significant” mainly referrers to statistical significance which is not the case in this context. Since the authors’ present main themes, this word can be replaced or avoided.
While quoting, the authors may consider to give details about participant – e.g. male, XX years old, from XX country, if possible duration of migration.
Some themes in the text are quoted several times. The authors may consider to select 1 quote that best describes each context (instead of several similar quotes).
Author Response
Thank you so much for providing a manuscript on this important issue. The manuscript would benefit from the revision.
Title: Resettlement in where? The authors could consider adding more clarity by indicating setting migrants from Africa who resettled in – Region/area/setting
Response 1: Thank you for your observation of omission of the setting in the title. The title has now been changed to:, “Telling Our Stories: Resilience during Resettlement for African Skilled Migrants in Australia”
Methods – the authors did not mention if the study follows Consolidated criteria for reporting qualitative research (COREQ): a 32-item checklist for interviews and focus groups – guidelines. The authors are suggested to carefully check COREQ and follow its principles thoughts the manuscript.
Response 1: Thank you for the suggestion. The methodology was guided by COREQ guidelines which has 32 items, and three Domains. The details about this guide have been added into the text.
Results –the authors may consider presenting main themes as tables.
Response 1: Given that only one reviewer out of three preferred to use the table in summarising the findings, and their comments in the point below about main themes being presented, we would like to conform to the two reviewers who did not indicate this as a preference, ie maintain thematic presentation in text supported by participants narrations. We believe that presenting themes supported by the participants narratives is a better mode of presentation, rather than putting these in a table. We recognise however that either way could be used, with the message being sustained.
Page 4, Line 177: the word “significant” mainly referrers to statistical significance which is not the case in this context. Since the authors’ present main themes, this word can be replaced or avoided.
Response 1: Thank you for the suggestions. We have made changes to the word “significance to align with the reviewers suggestions.
While quoting, the authors may consider to give details about participant – e.g. male, XX years old, from XX country, if possible duration of migration.
Response 1: Because the African community in Adelaide is very small, we have deliberately left these details out to improve privacy as promised in the statement to participants information. Some of the narratives are very personal a it will be very easy for the community members to identify these personal stories, should we not be careful in our presenting of information. Additionally, our Institutions’ ethics committees’ and our own experiences in this space suggest protection of privacy by making the findings as anonymous as possible.
Some themes in the text are quoted several times. The authors may consider to select 1 quote that best describes each context (instead of several similar quotes).
Response 1: Thank you for this observation. As the quotes were from different participants, we believe the effect is additive. There were not able to identify quotes that were repeated. Additionally, as stated above, participants had similar experiences and we have left out a large number of quotes which have similar meanings.